# Regulation of Inflammation-Related Genes through *Fosl1* Suppression in a Levetiracetam-Treated Pilocarpine-Induced Status Epilepticus Mouse Model

**DOI:** 10.3390/ijms23147608

**Published:** 2022-07-09

**Authors:** Rie Komori, Taira Matsuo, Aya Yokota-Nakatsuma, Ritsuka Hashimoto, Shizuka Kubo, Chihiro Kozawa, Tomomi Kono, Yasuhiro Ishihara, Kouichi Itoh

**Affiliations:** 1Laboratory for Pharmacotherapy and Experimental Neurology, Kagawa School of Pharmaceutical Sciences, Tokushima Bunri University, Tokushima 769-2193, Japan; komorir@kph.bunri-u.ac.jp (R.K.); t-matsuo@kph.bunri-u.ac.jp (T.M.); s138063@stu.bunri-u.ac.jp (R.H.); s138033@stu.bunri-u.ac.jp (S.K.); s148044@stu.bunri-u.ac.jp (C.K.); s148043@stu.bunri-u.ac.jp (T.K.); 2Laboratory of Immunology, Kagawa School of Pharmaceutical Sciences, Tokushima Bunri University, Tokushima 769-2193, Japan; yokotate@kph.bunri-u.ac.jp; 3Program of Biomedical Science, Graduate School of Integrated Sciences for Life, Hiroshima University, Hiroshima 739-8521, Japan; ishiyasu@hiroshima-u.ac.jp

**Keywords:** levetiracetam, *Fosl1*, inflammation, epileptogenesis, epilepsy after brain injury

## Abstract

Levetiracetam (LEV) suppresses the upregulation of proinflammatory molecules that occurs during epileptogenesis after status epilepticus (SE). Based on previous studies, LEV likely helps prevent the onset of epilepsy after insults to the brain, unlike other conventional anti-epileptic drugs. Recently, we discovered that the increase in *Fosl1* expression that occurs after lipopolysaccharide (LPS) stimulation is suppressed by LEV and that *Fosl1* inhibition suppresses inflammation in BV-2 microglial cells. These data indicate that *Fosl1* is an important target of LEV and a key factor in preventing epilepsy onset. In this study, we examined the effects of LEV on *Fosl1* expression and neuroinflammation in vivo. During epileptogenesis, the post-SE upregulation of hippocampal levels of *Fosl1* and many inflammatory factors were suppressed by LEV. *Fosl1* expression showed a characteristic pattern different from that of the expression of *Fos*, an immediate-early gene belonging to the same Fos family. At 2 days after SE, *Fosl1* was predominantly expressed in astrocytes but was rarely detected in microglia, whereas *Fos* expression was distributed in various brain cell types. The expression of A2 astrocyte markers was similar to that of *Fosl1* and was significantly suppressed by LEV. These results suggest that LEV may regulate astrocyte reactivity through regulation of *Fosl1*.

## 1. Introduction

Epilepsy is one of the most common serious neurological disorders. The incidence and prevalence of epilepsy worldwide are approximately 50 per 100,000 and 700 in 100,000 people per year, respectively, and are higher in infants (<2 years old) and older people [1,2,3]. Among them, epilepsy that develops after an insult to the brain, such as through stroke or brain trauma, often develops in the elderly and has become a major social problem in a society with an increasingly older population, so the prevention of these forms of epilepsy is an important area of focus. The key process underlying the onset of epilepsies arising from insults to the brain is thought to be the changes in the brain that occur during the early phase of epileptogenesis (latent period), and many drug discovery investigations for agents that inhibit these changes are underway. As a result, it has been found that some non-antiepileptic drugs (AEDs), unlike conventional AEDs, may prevent the onset of epilepsy [4,5,6,7,8,9]. It has also been suggested that the same effect could be seen with levetiracetam (LEV), one of the new AEDs [10,11,12].

LEV is an established second-generation AED that is widely used to treat partial onset and generalized seizures [13]. Our previous study showed that LEV treatment attenuated the spontaneous recurrent seizures and suppressed the blood−brain barrier (BBB) failure associated with the angiogenesis and brain inflammation that occur after pilocarpine (PILO)-induced status epilepticus (SE) [14,15,16]. The synaptic vesicle protein 2A (SV2A) membrane protein is known as the main target of action of LEV, but the mechanism by which it suppresses brain inflammation is unknown. As a result of efforts to explore new targets for LEV, we identified FOSL1 as another new target molecule of LEV in an in vitro experiment using murine BV-2 microglial cells [17]. It was shown that the microglial activation and inflammatory response that occur after lipopolysaccharide (LPS) stimulation were inhibited through the suppression of FOSL1 expression that was induced by LEV administration. FOSL1 belongs to the FOS family and binds a Jun-family member to form the transcription factor AP-1, which is involved in a variety of biological processes, including cell proliferation, differentiation, apoptosis, and inflammation [18,19,20].

In the present study, we comprehensively investigated gene expression in the hippocampus of LEV-treated and untreated PILO-SE mice to promote understanding of the whole process of early epileptogenesis. Additionally, to investigate whether LEV suppresses neuroinflammation after SE by controlling the activation of microglia via *Fosl1* as well as in vitro, the dynamics of hippocampal *Fosl1* expression were investigated in detail, and the possibility that LEV administration can suppress epileptogenicity by regulating *Fosl1* expression was explored.

## 2. Results

### 2.1. Comprehensive Analysis of Hippocampal Gene Expression Profiling during Epileptogenesis after PILO-SE

Epilepsy occurring after insults to the brain develops in three phases: injury (brain insult), epileptogenesis (latent period) and subsequent chronic epilepsy (spontaneous recurrent seizures). We have previously reported that levetiracetam (LEV) could be involved in neuroprotection via anti-angiogenesis and anti-inflammatory activities against the blood−brain barrier (BBB) dysfunction that occurs in the acute phase of epileptogenesis after status epilepticus (SE) [15], but the mechanism remains unclear. To comprehensively and effectively assess changes in gene expression in this early stage, we applied cap analysis of gene expression (CAGE) to the hippocampus of PILO-SE mice that were treated with LEV or untreated at 6 h and 2 days after SE. To explore the potential mechanism of LEV action, we used Gene Ontology (GO) analysis of biological processes and identified the group of genes that exhibited upregulated levels after PILO-SE and for which this upregulation was prevented by LEV treatment (Figure 1A). The levels of genes involved in blood vessel morphogenesis, cell migration and inflammatory response were upregulated at 6 h after SE, which is the earlier stage of epileptogenesis (Figure 1A, left upper panel). The expression levels of genes related to immunity and inflammation were significantly increased at 2 days after SE, when the stage was more advanced (Figure 1A, right upper panel). At either stage, the increases in the expression levels of genes involved in inflammation were prevented by LEV administration (Figure 1A, lower panels). Figure 1B summarizes the results of the analysis of the changes in the gene expression levels of factors that commonly influence BBB function (Figure 1B, upper panels) and cytokines/chemokines (Figure 1B, lower panels) at 6 h and 2 days after SE. The expression levels of these factors were often elevated after SE (Figure 1B, open circles), but these changes were found to be suppressed by LEV administration (Figure 1B, filled circles). For the levels of many BBB permeability-related genes, increased expression was already observed 6 h after SE. In particular, the increased expression levels of *Cdh5* (vascular endothelial cadherin, VE-cadherin), which is selectively expressed in vascular endothelial cells and is a useful marker of BBB function, was remarkable. The increase in expression was induced even at 2 days after SE, but the difference was smaller than that observed after 6 h. However, the increased levels of cytokine/chemokine expression persisted even 2 days after SE. These observations were in excellent agreement with our previous magnetic resonance imaging (MRI) studies showing that LEV treatment prevented the BBB failure associated with the brain inflammation and angiogenesis that occurred within 2 days of SE [15]. To confirm the CAGE results and examine the expression levels of cytokines/chemokines more closely, total RNA was extracted from the hippocampi of control animals (pre-SE) and PILO-SE (post-SE) mice that were treated with LEV or untreated, and the RNA was analyzed using real-time PCR (Figure 2). The gene expression levels of many proinflammatory cytokines/chemokines, such as *Il6*, *Ccl2* and *Cxcl1*, were significantly elevated at 2 days after SE, and upregulation of their expression was prevented by treatment with LEV; thus, results similar to those of CAGE were obtained with real-time PCR.

Moreover, to assess the protein expression levels of specific inflammation-related molecules, a Proteome Profiler Mouse XL Cytokine Array Kit was used (Figure 3). Of the 111 proteins with expression levels that can be assessed using this kit, 48 proteins showed significantly higher expression levels at 2 days after SE than pre-SE (Figure 3A, middle panel). Among them, the levels of 23 proteins, including CCL6, CXCL10 and CD14, were not upregulated after SE following LEV administration (Figure 3A, lower panel). The fluctuations in the expression levels of typical cytokines/chemokines are shown in Figure 3B. LEV administration was shown to have a suppressive effect on cytokine/chemokine production at both the mRNA and protein levels.

### 2.2. Induction of Hippocampal Fosl1 Expression during Epileptogenesis after PILO-SE and Differences between Fosl1 and Fos Expression

SE is associated with the excessive synaptic activation of excitatory neuronal circuits [21]. Neural activity has been shown to induce the expression of immediate early genes (IEGs) encoding effector molecules and transcription factors [22,23]. Rapidly and transiently induced IEGs cause changes in response to stimuli through different functions. To examine the dynamics of IEGs in the hippocampus during epileptogenesis, the expression patterns of major neuronal IEGs were analyzed using CAGE data (Figure 4A). Many IEGs were observed to exhibit a rapid increase in expression after 6 h of SE, and it was shown that high levels were maintained even after 2 days (Figure 4A, open circles). In addition, for most factors, this induced increase in expression was suppressed by LEV administration (Figure 4A, filled circles).

To explore the mechanism underlying the suppression of the inflammatory reaction that occurs after LEV administration, we focused on the expression profiles of *Fosl1* in the hippocampus after SE. *Fosl1* (also known as *Fra1*), a marker for neuronal activation, is a member of the Fos gene family that plays important roles in complex cellular processes, such as differentiation, proliferation and apoptosis. In our previous study using murine BV-2 microglial cells, we identified *Fosl1* as a new target of LEV by CAGE and RNA interference (RNAi) [17]. We have already mentioned that an increase in the expression of *Fosl1* in the hippocampus was induced during epileptogenesis in a mouse model of PILO-SE, and that the levels were reduced by consecutive administration of LEV immediately after SE. The CAGE data collected in the current study also showed differences in hippocampal *Fosl1* expression levels (higher than those in pre-SE mice; logFC = 8.77, *p* < 0.01 at 6 h, logFC = 9.07, *p* < 0.01 at 2 days after SE) and the effects of LEV administration (lower than those in untreated mice; logFC = −2.23, *p* < 0.01 at 6 h, logFC = −1.94, *p* < 0.05 at 2 days after SE) (Figure 4A). To compare the detailed dynamics of the levels of *Fosl1* and other members of the Fos family, total RNA was extracted from the hippocampi of naive (pre-SE) mice, mice at the midway stage of SE (after the first, third and fifth generalized convulsive seizures) or post-SE (1 h, 3 h, 2 days, or 7 days) mice and analyzed using real-time PCR (Figure 4B). The increased expression of *Fos*, which is well known as one of the neuronal IEGs, was induced immediately after treatment with PILO, with a peak at 1 h post-SE (approximately 70-fold higher than pre-SE levels). This induction was transient and nearly disappeared within 24 h post-SE. However, *Fosl1* expression was rarely detected during the convulsive seizures (CSs) and was induced only after 1 h of SE, with a peak of 1 to 2 days post-SE (approximately 140-fold higher than pre-SE levels). These results indicated that both hippocampal *Fosl1* and *Fos* expression were induced during epileptogenesis, but *Fosl1* has a completely different expression pattern from that of *Fos*; nevertheless, they belong to the same Fos family. *Fos* expression began to be induced immediately after even one CS, and its expression levels decreased rapidly after the end of the CSs, whereas *Fosl1* expression began to be induced during the post-SE phase (Figure 4B). The high expression of *Fosl1* tended to persist into the late stages of epileptogenesis, suggesting that *Fosl1* was more closely involved in the progression of epileptogenesis than in the seizures themselves.

We explored whether LEV administration shows different effects on the induction of *Fosl1* and *Fos* gene expression after PILO-SE. As reported previously, LEV treatment tended to reduce *Fosl1* expression 1 h post-SE and significantly reduced *Fosl1* expression 1 and 2 days post-SE (Figure 4C, left panel). A similar suppressive effect of LEV on the increase in *Fos* levels was observed at 1 h post-SE (Figure 4C, right panel). These findings suggested that LEV treatment has a suppressive effect on hippocampal *Fosl1* and *Fos* mRNA expression after SE.

### 2.3. Identification of Cell Types Expressing Fosl1 in the Mouse Hippocampus after SE

In our previous study, we reported that prevention of *Fosl1* upregulation by LEV suppressed microglial activation in murine BV-2 microglial cells treated with LPS [17]. To determine the cell types in which *Fosl1* is expressed during epileptogenesis after PILO-SE, we examined the expression pattern of *Fosl1* in neurons, glial cells (microglia and astrocytes) and vascular endothelial cells separated using a FACSAria II (Figure 5). CD11b^+^, CD90.2^+^, EAAT2^+^ and CD144^+^ cells were separated from the hippocampi of naive (pre-SE) and post-SE mice treated with LEV or untreated (2 days post-SE) (Figure 5A), and the transcript expression levels of *Fosl1* and *Fos* were analyzed. To ascertain whether the isolation criteria were appropriate, we also monitored the expression of some marker genes of specific brain cell types, such as *Tmem119* (microglia-specific marker), *Grin1* (neuron-specific marker), *Slc1a2* (astrocyte-specific marker) and *Cdh5* (vascular endothelial cell-specific marker), in these four populations using real-time PCR (Figure 5B). The CD11b^+^ cells expressed high levels of *Tmem119* (Figure 5B, left upper panel), confirming that they were microglia. The CD90.2^+^ cells expressed high levels of *Grin1* (Figure 5B, right upper panel), confirming that they were neurons. The EAAT2^+^ cells expressed high levels of *Slc1a2* (Figure 5B, left lower panel), confirming that they were astrocytes. The CD144^+^ cells expressed high levels of *Cdh5* (Figure 5B, right lower panel), confirming that they were vascular endothelial cells. The expression levels of *Tmem119* in CD11b^+^ cells, *Grin1* in CD90.2^+^ cells and *Slc1a2* in EAAT2^+^ cells were significantly decreased at 2 days post-SE. These reductions in *Tmem119* in CD11b^+^ cells and *Slc1a2* in EAAT2^+^ cells were significantly suppressed by LEV administration. There was no significant difference in the levels of *Grin1* in CD90.2^+^ cells between the LEV-treated group and the untreated group, but a recovery tendency was observed. These results indicated that LEV might be effective in preventing the decline in neuronal and glial cell function caused by PILO-SE.

Under such conditions, *Fosl1* mRNA levels were very low in CD11b^+^ cells, CD90.2^+^ cells and CD144^+^ cells (Figure 5C, left panel). The highest level of expression was observed in EAAT2^+^ cells (namely, astrocyte populations). Although there was no significant difference, LEV treatment tended to reduce this upregulation in *Fosl1* expression at 2 days post-SE. However, *Fos* expression was evenly distributed in most major brain cell types (Figure 5C, right panel). These results indicated that hippocampal *Fosl1* expression was limited to astrocytes at 2 days post-SE, despite a wide range of *Fos* expression levels.

### 2.4. Effect of PILO-SE and LEV Administration on the Induction of Different Reactive Astrocyte Phenotypes

Astrocytes are key players in the regulation of brain tissue homeostasis and neuronal excitability [24], and many reports show that astrocytes contribute to the pathophysiology of epilepsy [25,26,27,28]. To examine the effect of SE and LEV administration on astrocytes, total RNA was extracted from the hippocampi of pre-SE and post-SE mice treated with LEV or that were untreated, and the levels of genes specifically expressed in astrocytes were analyzed using real-time PCR (Figure 6A). The increased expression of *Gfap*, the most commonly used astrocyte marker, was induced within 1 day post-SE, and the levels remained high for up to 7 days (Figure 6A, left upper panel). Although the effect was not significant, LEV administration tended to reduce this upregulated expression level for at least 1 day after treatment. No significant differences were found in the levels of other markers, such as *P2ry1*, *Ndrg2* and *Slc1a2*. Since *Gfap* is highly expressed by reactive astrocytes [29,30], the findings suggest that epileptogenesis promotes the presence of reactive astrocytes.

Recent studies have revealed that there are two different reactive astrocyte populations in the adult central nervous system (CNS): proinflammatory or neurotoxic A1 astrocytes and anti-inflammatory or neuroprotective A2 astrocytes [30,31,32,33]. To examine the effects of SE and LEV administration on the induction of A1 and A2 astrocyte phenotypes, detailed time-courses of the expression of A1-specific markers (*C3*, *H2-D1*, *Ggta1*, *Gbp2* and *Amigo2*) (Figure 6B) and A2-specific markers (*Ptx3*, *S100a10* and *Ptgs2*) (Figure 6C) in the hippocampus during epileptogenesis were assessed using real-time PCR. The transcript levels of *C3* and *H2-D1* began to increase within approximately 3 h post-SE and continued to increase until 7 days. *Ggta1* mRNA levels reached a plateau after 1 day and remained at high levels until 7 days. The expression of *Gbp2* also reached high levels after 1 day but decreased shortly thereafter. The expression of *Amigo2* tended to decrease gradually over the second day, but no significant difference was observed between pre-SE and post-SE. In either case, no significant differences were observed between the LEV-treated group and the untreated group, suggesting that the presence of some A1-phenotype astrocytes may have been induced after SE and that these astrocytes were not affected by LEV administration.

The expression levels of the A2-specific markers *Ptx3* and *S100a10* were significantly increased 1 and 2 days after SE, and these increases were suppressed by administration of LEV (Figure 6C). The mRNA level of *Ptgs2*, another A2 marker, was dramatically increased within a few hours but returned to low levels within 1 to 2 days. No effect of LEV administration on this change was observed. These data indicated that the presence of both A1- and A2-phenotype astrocytes increased after SE, and LEV administration had an effect on the increase in the presence of the A2-phenotype rather than the presence of the A1-phenotype.

## 3. Discussion

Levetiracetam (LEV), which has unique pharmacokinetic properties and a specific mode of action and chemical structure, is one of the most widely used second-generation anti-epileptic drugs (AEDs) for both adults and children [13,34]. The synaptic vesicle protein 2A (SV2A) membrane protein is known as one of the main target factors for LEV. The existence of other mechanisms of action has been suggested, but these mechanisms have been unknown for a long time. Recently, *Fosl1* was newly identified as a target molecule for LEV in an in vitro experimental system using murine BV-2 microglial cells [17]. A mechanism was shown in which LEV inhibited microglial activation and the inflammatory response by suppressing the increase in *Fosl1* expression that occurs after LPS treatment. In this study, to clarify the mechanism of action of LEV in vivo, we conducted a comprehensive gene expression analysis (Figure 1) and investigated the dynamics of *Fosl1* expression, which fluctuated in vitro (Figure 4). The results indicated that the expression levels of many cytokines/chemokines were dramatically upregulated along with the expression level of *Fosl1* during epileptogenesis after SE, and this upregulation was suppressed by LEV administration (Figure 2 and Figure 3).

Cytokine storms, or cytokine cascades, which are associated with a wide variety of infectious and noninfectious diseases, are caused by the excessive production of proinflammatory cytokines [35,36]. After SE, it was shown that proinflammatory cytokines were intensely released and cytokine storm-like phenomena were present in the hippocampus, so suppressing inflammation as soon as possible may be an effective strategy for reducing the development of subsequent post-SE epilepsy. Although various clinical trials have tested the effect of conventional AEDs, such as phenobarbital, valproate, carbamazepine, phenytoin, lamotrigine and topiramate, on the prevention of the onset of epilepsy, none of them were effective [4,5,6,9,15]. LEV has been suggested to have the potential to reduce the risk of acquired epilepsy or prevent the development of epilepsy [10,11,12], but the mechanism underlying these effects is unknown. Since the cytokine storm-like phenomenon after SE was suppressed by LEV administration, it has been suggested that LEV affected not the expression of individual inflammatory cytokines but that of the upstream transcription factors, such as IEGs and NF-κB, and was involved in the control of the inflammatory state in the brain.

In vivo experiments at 2 days post-SE revealed that hippocampal *Fosl1* expression was hardly observed in microglia and was mainly found in astrocytes (Figure 5). Since LEV is known to suppress neuroinflammation and abnormal microglial activation [15,16], a target search using BV-2 microglial cells was performed, and *Fosl1* was identified [17]. It is predicted that LEV regulates microglial activation via *Fosl1* to suppress neuroinflammation after SE, but the inflammatory response might be suppressed through regulation of astrocyte activity rather than microglia.

The difference in the induction of *Fosl1* and *Fos* gene expression after SE was also clarified. *Fos* is known to be involved in neuronal survival and to induce neuronal cell death [37,38]. Given this information, we consider that *Fos*, whose expression is induced first after SE, triggers neuronal cell death, while *Fosl1*, whose expression is induced thereafter, causes neuroinflammation via astrocyte activation, further increasing neuronal cell death. LEV administration suppresses the induction of these factors. Notably, neuronal cell death has been shown to occur in the hippocampal CA1 at 2 days after SE, but this effect is prevented by LEV treatment [15].

It has been reported that decreased expression of microglial homeostatic genes, such as *Tmem119*, correlates with the severity of neurodegeneration and that decreased expression levels of homeostatic microglial markers might be hallmarks of progressive neuronal loss [39]. Similarly, neurodegeneration could lead to a decline in the expression of neuronally regulated genes, such as *Slc1a2*, in astrocytes [40]. Since decreases in the expression of these markers were confirmed at 2 days post-SE in this study as well (Figure 5), it was found that neurodegeneration was induced by SE and that LEV administration suppressed the progression of neurodegenerative disorder.

Glial cells are nonneuronal cells in the CNS that can be classified into three major groups: astrocytes, microglia and oligodendrocytes [41,42]. Astrocytes are the most abundant glial cell population in the CNS. For many years, astrocytes were thought to provide only metabolic and physical support for neurons [27]. However, in recent years, it has become clear that astrocytes also play an essential role in brain function, such as in the efficiency of synaptic transmission. Moreover, it has become clear that reactive astrocytes are further classified into A1 (neurotoxic) and A2 (neurorestorative) phenotypes according to their functions [31]. A1 reactive astrocytes can secrete neurotoxins that induce neural cell death, while A2 reactive astrocytes promote neuronal survival and tissue repair after nerve injury in a variety of neurological diseases [43,44,45]. Astrocytes are important regulators of the inflammatory response involved in the rapid recovery of nervous tissue [46].

In this study, LEV administration after SE significantly reduced the presence of A2-phenotype astrocytes (Figure 6). The decrease in the presence of the anti-inflammatory A2-phenotype suggests that levels of inflammation have been rapidly reduced. Since the increase in *Fosl1* expression and the increase in the levels of some A2-phenotype markers after SE were very similar, it was possible that *Fosl1* was involved in the induction of the presence of A2-phenotype astrocytes. Although there was no significant effect of LEV administration on the presence of A1-phenotype astrocytes, it seems to have the effect of suppressing the increase in the levels of A1 astrocytes to some extent. Because microglial activation is required for the production of A1-phenotype astrocytes [31,47], if *Fosl1* is also involved in the inhibition of microglial activation that occurs earlier than the second day after SE, similar to the results obtained in vitro, it is considered that LEV administration suppresses the increase in *Fosl1* expression, which reduces the reactivity of microglia and astrocytes, the levels of the inflammatory reaction and the occurrence of consequential neuronal cell death. This study revealed changes in hippocampal *Fosl1* expression, suggesting that LEV might inhibit the onset of spontaneous convulsions by suppressing the levels of intracerebral inflammation via *Fosl1* regulation in glial cells after SE.

It has been suggested that the transcription of *Fosl1* is regulated by other transcription factors that work further upstream, such as *Egr1* [48,49]. Although there was a possibility that the expression of hippocampal *Fosl1* after SE might be regulated by *Egr1*, the increased expression of *Egr1* observed after SE was not affected by LEV administration in the experimental system used in this study (Appendix A). It was suggested that LEV directly controlled *Fosl1* expression without affecting *Egr1* expression. However, there is still the possibility that many other factors are involved, so further investigations are underway to clarify the upstream regulatory factors controlling *Fosl1* using the CAGE data.

## 4. Methods and Materials

### 4.1. Experimental Animals

All animal experiments were approved by the Tokushima Bunri University Animal Care Committees and were performed in accordance with the National Institutes of Health (USA) Animal Care and Use Protocol. All efforts were made to minimize the number of animals used and their suffering. Male, eight-week-old ICR (CD-1) mice were purchased from Japan SLC (Shizuoka, Japan). All mice were maintained with laboratory chow and water ad libitum on a 12 h light/dark cycle. The utilized animals were euthanized using saturated KCl.

### 4.2. Induction of Status Epilepticus (SE) by Pilocarpine (PILO) and Administration of Levetiracetam (LEV)

The PILO-induced SE model was established according to our previous description [14]. Briefly, ICR mice (9–10 weeks old) were injected with methyl scopolamine (1 mg/kg, Sigma-Aldrich, St. Louis, MO, USA) intraperitoneally (i.p.) 30 min prior to PILO (290 mg/kg, i.p., Sigma-Aldrich, St. Louis, MO, USA) administration. The animals were placed in a plastic chamber (10 × 15 × 30 cm), and their behavior was observed before and after PILO administration. The control mice were injected with saline (0.1 mL/10 g i.p.) instead of PILO. SE was defined by the occurrence of five generalized convulsive seizures. To terminate SE, all mice were injected with diazepam (DZP; Cercine^®^, 10 mg/kg, i.p.; Takeda Pharmaceutical Ltd. Osaka, Japan), once or more as needed to suppress convulsive seizures. LEV (LKT Labs, Inc., St. Paul, MN, USA) was orally administered at a dose of 360 mg/kg within 30 min after DZP injection and thereafter twice a day (at 8:30 and 17:30). As a result, the SE + LEV 6 h group was given LEV once, and the SE + LEV 2 days group was given LEV 5 times.

### 4.3. CAGE Analysis

Total RNA was extracted from the whole mouse hippocampus using a High Pure RNA Isolation Kit (Roche Diagnostics K.K., Tokyo, Japan). The RNA quality check, CAGE library preparation, sequencing, mapping and data analysis for CAGE were performed by DNAFORM (Yokohama, Kanagawa, Japan). In brief, the quality of total RNA was assessed with a Bioanalyzer (Agilent Technologies, Santa Clara, CA, USA) to ensure that the RNA integrity number (RIN) was over 7.0. cDNA was synthesized from the total RNA using random primers. CAGE libraries were constructed and sequenced using single end reads of 75nt on a NextSeq 500 instrument (Illumina, San Diego, CA, USA). The obtained reads (CAGE tags) were mapped to the mouse mm9 genome using BWA (version 0.7.17). The unmapped reads were then mapped with HISAT2 (version 2.0.5). CAGE tag clustering, differentially expressed gene detection and motif discovery were performed with the RECLU pipeline. Tag count data were clustered using the modified Paraclu program. Clusters with counts per million (CPM) values < 0.1 were discarded. Regions with 90% overlap between replicates were extracted by BEDtools (version 2.12.0). The clusters with irreproducible discovery rate (IDR) values ≥ 0.1 and clusters longer than 200 bp were discarded. Differentially expressed genes were detected using the edgeR package (version 3.22.5). The list of differentially expressed genes detected by RECLU with a false discovery rate (FDR) ≤ 0.05 were subjected to GO enrichment analysis with the clusterProfiler package. The raw data were registered in the NCBI GEO database under accession number GSE205373.

### 4.4. Real-Time Polymerase Chain Reaction (PCR) Analysis

Total RNA was isolated using a High Pure RNA Isolation Kit (Roche Diagnostics K.K., Tokyo, Japan) for the whole mouse hippocampus and NucleoSpin RNA XS (Macherey-Nagel, Düren, Germany) for FACS-sorted cells. The concentration and quality of the eluted RNA were measured by UV absorbance at 260 nm and 280 nm using a NanoDrop Spectrophotometer (Nanodrop Technologies, Wilmington, DE, USA). cDNA was synthesized using random primers, a ReverTra Ace reverse transcriptase (TOYOBO Co., Osaka, Japan) and a PCR Thermal Cycler (Takara Biomedicals, Tokyo, Japan) according to the manufacturers’ instructions. Gene-specific primers, cDNA template and SYBR Green PCR Master Mix (Thermo Fisher Scientific, Waltham, MA, USA) were used to PCR amplify target genes; amplification was run for 40 cycles using a QuantStudio 7 Flex Real-Time PCR System (Thermo Fisher Scientific, Waltham, MA, USA). The instrument’s dissociation protocol was used to verify that only the specified PCR products were detected. The relative amounts of target transcripts were calculated using a standard curve generated with serial cDNA dilutions, and the levels were normalized to that of β-actin (*Actb*) within the same cDNA sample. The primer sequences used in this study are listed in Table 1.

### 4.5. Mouse Proteomic Profiling Array Analysis

A membrane-based antibody array kit (Proteome Profiler Mouse XL Cytokine Array Kit, ARY028 R&D Systems, Inc., Minneapolis, MN, USA) coated with 111 capturing antibodies in duplicate on a nitrocellulose membrane was used to determine the relative levels of selected mouse cytokines, chemokines and growth factors. Hippocampal homogenates were prepared from mice before SE (pre-SE) and 2 days post-SE (SE 2 days) that were treated with LEV or were untreated, and the homogenates were tested according to the manufacturer’s instructions. Arrays were imaged using a LAS-3000 (Fujifilm, Tokyo, Japan), and images were analyzed using Multi Gauge software (Fujifilm, Tokyo, Japan). Three independent experiments were performed.

### 4.6. FACS of Microglia, Astrocytes, Neurons and Vascular Endothelial Cells

Mice were perfused transcardially with PBS while under deep anesthesia. The brains were then removed, and the hippocampus was dissected. The hippocampus was digested using collagenase (400 U/mL) and an Adult Brain Dissociation Kit (Miltenyi Biotec, Pris, France). After filtering with a 100 µm cell strainer (Becton Dickinson Biosciences, San Jose, CA, USA), the suspended cells were centrifuged at 400× *g* for 10 min, and the pelleted cells were resuspended in Cell Staining Buffer (BioLegend, San Diego, CA, USA). The dissociated single-cell suspensions were blocked with anti-mouse CD16/CD32 against Fc receptors (Becton Dickinson Biosciences, San Jose, CA, USA) for 10 min on ice. Dead cells were stained with propidium iodide (PI). The cell suspensions were incubated and costained for 30 min on ice in the dark with FITC-conjugated anti-mouse/human CD11b (BioLegend, San Diego, CA, USA, 101206), PE-conjugated anti-mouse CD90.2 (BioLegend, San Diego, CA, USA, 105307), Alexa Fluor 647-conjugated anti-EAAT2 (Signalway Antibody LLC, Greenbelt, MD, USA, #C45336-AF647) and Brilliant Violet 421-conjugated anti-mouse CD144 (BioLegend, San Diego, CA, USA, 138013). The excess unbound antibody was then removed by washing with PBS. Stained cell suspensions were prefiltered through a 0.22 µm pore size membrane and sorted by a FACSAria II cell sorter (Becton Dickinson Biosciences, San Jose, CA, USA). FlowJo software was used for flow cytometry data analysis. Sorted cells were centrifuged at 400× *g* for 15 min, and pellets were suspended for RNA extraction using the NucleoSpin RNA XS (Macherey-Nagel, Düren, Germany).

### 4.7. Statistical Analysis

All data are presented as the mean ± standard error of the mean (SEM). Data were analyzed for significant differences by one-way ANOVA and Tukey’s post-hoc test. Results with *p* values of < 0.05 were considered statistically significant.

## 5. Conclusions

We performed a comprehensive gene expression analysis and showed that cytokine storm-like phenomena occur in the hippocampus after PILO-SE. It was suggested that LEV suppressed this phenomenon through the regulation of *Fosl1* levels expressed in hippocampal astrocytes. This may be one of the important factors explaining the preventive effect of LEV on the onset of epilepsy occurring after insults to the brain.

## Figures and Tables

**Figure 1 ijms-23-07608-f001:**
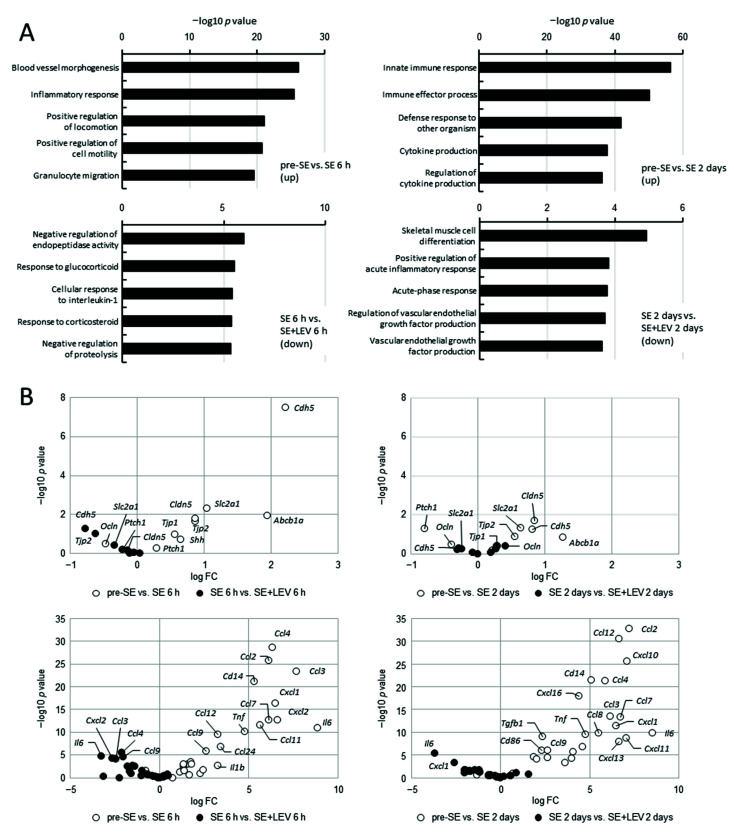
Comprehensive analysis of variable-expression genes after PILO-SE. (**A**) Functional annotation of upregulated genes in the hippocampus of PILO-SE mice (upper panels) and those with upregulation that was prevented after LEV treatment (lower panels) at 6 h and 2 days after SE. Total RNA was extracted from the hippocampi of mice treated with PILO after 6 h and 2 days (post-SE) or before SE (pre-SE), and cap analysis of gene expression (CAGE) and gene ontology (GO) enrichment analysis were performed. Two mice were used in each group. The top five GO terms in each group are displayed. The *x*-axis shows the −log10 *p* value, such that a higher value indicates greater significance. (**B**) Volcano plot showing differences in the expression levels of typical factors that influence BBB function (upper panels) and cytokine/chemokine (lower panels) between pre- and post-SE (open circle) or in animals treated with LEV or untreated (filled circle) at 6 h (left panels) and 2 days (right panels) after SE. The *x*-axis shows the fold change values (log2) between two experimental groups, and the *y*-axis shows the −log10 *p* value.

**Figure 2 ijms-23-07608-f002:**
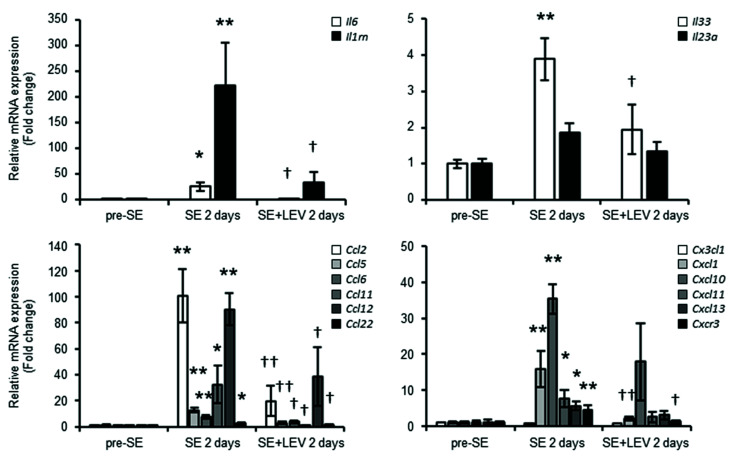
Real-time PCR analysis of the levels of typical cytokines/chemokines at 2 days after SE. Total RNA was extracted from the hippocampus before (pre-SE) and 2 days after SE (SE 2 days) in animals treated with LEV or that were untreated. Real-time PCR was performed using gene-specific primers as described in Table 1. All data points represent the mean ± SEM of at least three independent experiments (*n* = 3–9). Data were analyzed for significant differences by one-way ANOVA and Tukey’s post-hoc test. Results with *p* values < 0.05 were considered statistically significant. ** *p* < 0.01, * *p* < 0.05 vs. pre-SE mice, †† *p* < 0.01, † *p* < 0.05 vs. without post-SE mice.

**Figure 3 ijms-23-07608-f003:**
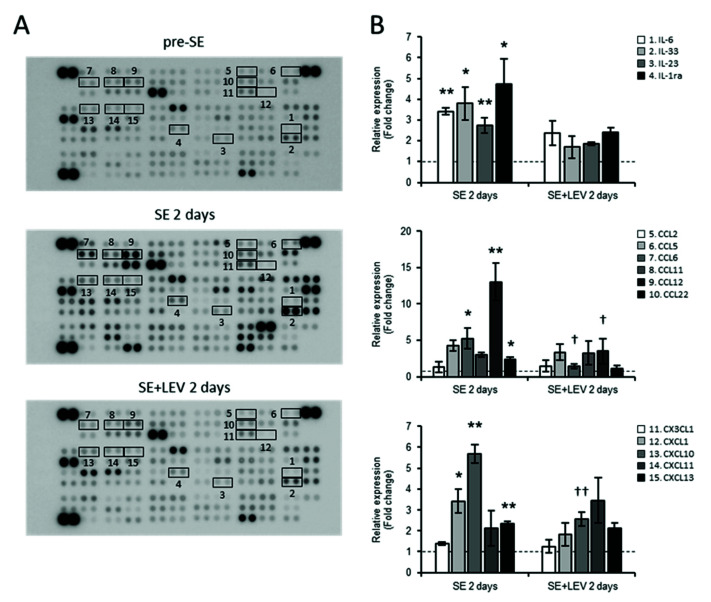
Alterations in the protein expression of hippocampal cytokines/chemokines after PILO-SE in mice treated with LEV or untreated. Protein extracts from the hippocampi of LEV-treated mice and untreated PILO-SE mice at 2 days post-SE or pre-SE were analyzed using the Proteome Profiler Array, Mouse XL Cytokine Array Kit (ARY028 R&D System), which contained 111 different capture antibodies spotted in duplicate on a nitrocellulose membrane. (**A**) Representative images of array membranes corresponding to pre-SE (upper panel), SE 2 days (middle panel) and SE + LEV 2 days (lower panel). Fifteen factors surrounded by squares were selected as representative cytokines/chemokines, and the quantified values at 2 days post-SE divided by those of the pre-SE are shown in (**B**). All data points represent the mean ± SEM of three independent experiments (*n* = 3). Data were analyzed for significant differences by one-way ANOVA and Tukey’s post-hoc test. Results with *p* values < 0.05 were considered statistically significant. ** *p* < 0.01, * *p* < 0.05 vs. pre-SE mice, †† *p* < 0.01, † *p* < 0.05 vs. without post-SE mice.

**Figure 4 ijms-23-07608-f004:**
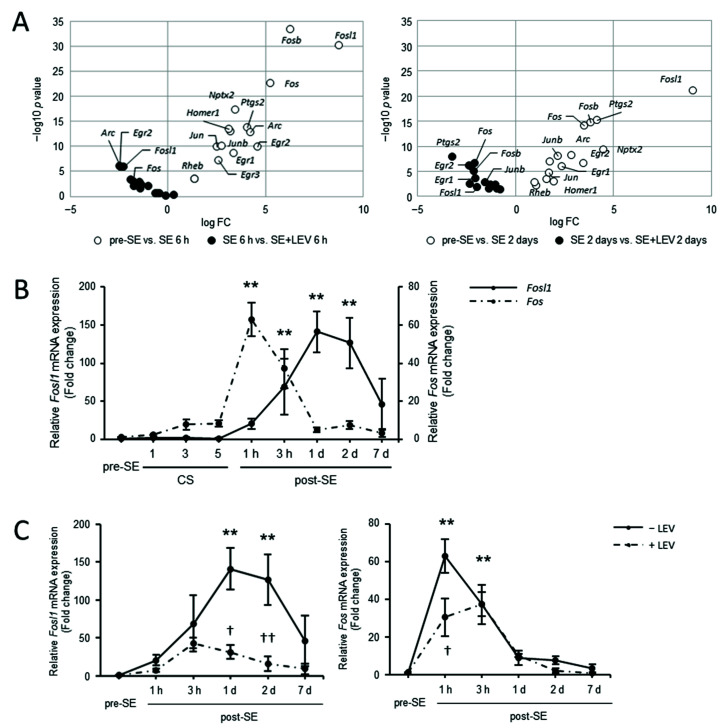
Alterations in gene expression levels of hippocampal neuronal IEGs after PILO-SE in mice treated with LEV or untreated. (**A**) Volcano plot showing major neuronal IEG expression differences between pre- and post-SE (open circles) or in mice treated with LEV or untreated (filled circles) at 6 h (left panel) and 2 days (right panel) after SE. The *x*-axis shows the fold change values (log2) between two experimental groups, and the *y*-axis shows the −log10 *p* value. (**B**) Time-course real-time PCR analysis of *Fosl1* and *Fos* mRNA expression in the hippocampus after PILO-SE. Total RNA was extracted from the hippocampi of mice treated with PILO after the first, third and fifth consecutive generalized convulsive seizures (CSs); at 1 h, 3 h, 1 day, 2 days and 7 days post-SE; and pre-SE. Real-time PCR was performed using gene-specific primers as described in Table 1. (**C**) Real-time PCR analysis of *Fosl1* and *Fos* mRNA expression in the hippocampus after SE in mice treated with LEV or untreated. Total RNA was extracted from the hippocampi of mice treated with PILO at 1 h, 3 h, 1 day, 2 days and 7 days post-SE and pre-SE in mice treated with LEV or untreated. The values were normalized relative to the β-actin (*Actb*) mRNA level and then divided by those of the control group (pre-SE) to calculate the relative mRNA levels. All data points represent the mean ± SEM of at least three independent experiments (*n* = 3–9). Data were analyzed for significant differences by one-way ANOVA and Tukey’s post-hoc test. Results with *p* values < 0.05 were considered statistically significant. ** *p* < 0.01 vs. pre-SE mice, †† *p* < 0.01, † *p* < 0.05 vs. without post-SE mice.

**Figure 5 ijms-23-07608-f005:**
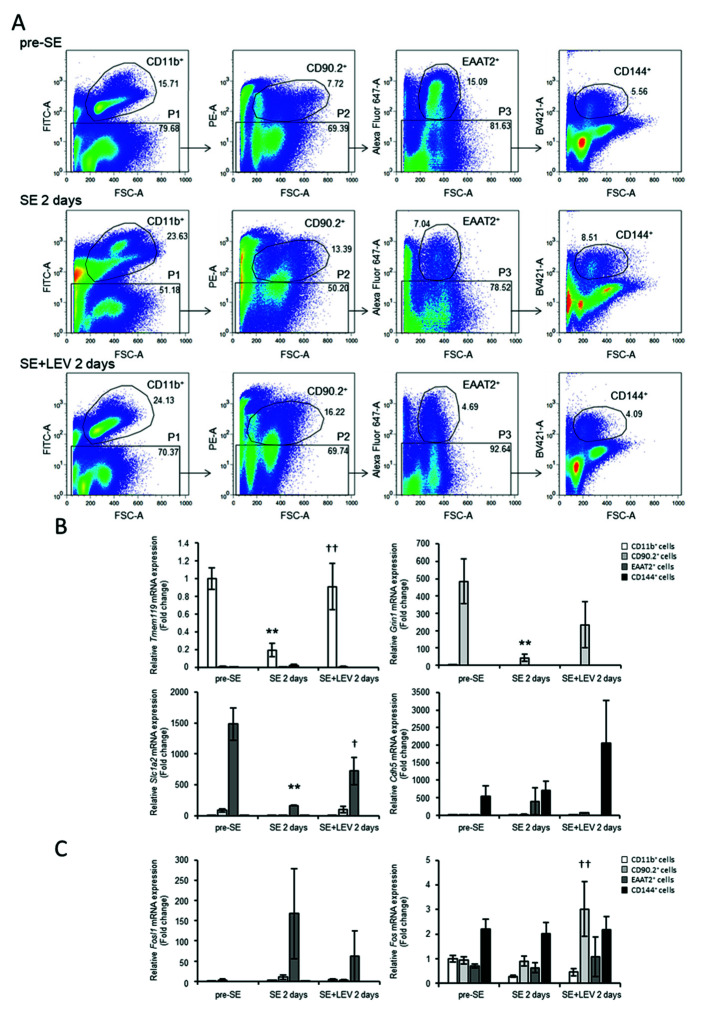
Identification of hippocampal cells expressing *Fosl1* after PILO-SE. (**A**) Representative plots of fluorescence-activated cell sorting from the hippocampus. The hippocampus was collected from PILO-SE mice at 2 days post-SE in mice treated with LEV or untreated and from pre-SE mice. Cells prepared for FACS analysis were first gated based on CD11b-FITC fluorescence intensity as CD11b^+^ or CD11b^−^. Next, CD11b^−^ cells (P1) were resolved according to CD90.2-PE fluorescence. CD90.2^−^ cells (P2) were further divided as EAAT2^+^ or EAAT2^−^ according to Alexa Fluor 647 fluorescence, and EAAT2^−^ cells (P3) were finally subdivided as CD144^+^ or CD144^−^ according to CD144-BV421 fluorescence. These four populations (CD11b^+^, CD90.2^+^, EAAT2^+^ and CD144^+^) were collected by FACS for analysis of transcript expression levels using real-time PCR. (**B**) Real-time PCR analysis of gene markers of microglia (*Tmem119*), neurons (*Grin1*), astrocytes (*Slc1a2*) and vascular endothelial cells (*Cdh5*) of four FACS-sorted populations. (**C**) Real-time PCR analysis of *Fosl1* and *Fos* expression levels of four FACS-sorted populations. The values were normalized relative to the β-actin (*Actb*) mRNA level and then divided by those of the control group (CD11b^+^ cells of pre-SE) to calculate the relative mRNA levels. All data points represent the mean ± SEM of at least three independent experiments (*n* = 3–7). Data were analyzed for significant differences by one-way ANOVA and Tukey’s post-hoc test. Results with *p* values < 0.05 were considered statistically significant. ** *p* < 0.01 vs. pre-SE mice, †† *p* < 0.01, † *p* < 0.05 vs. without post-SE mice.

**Figure 6 ijms-23-07608-f006:**
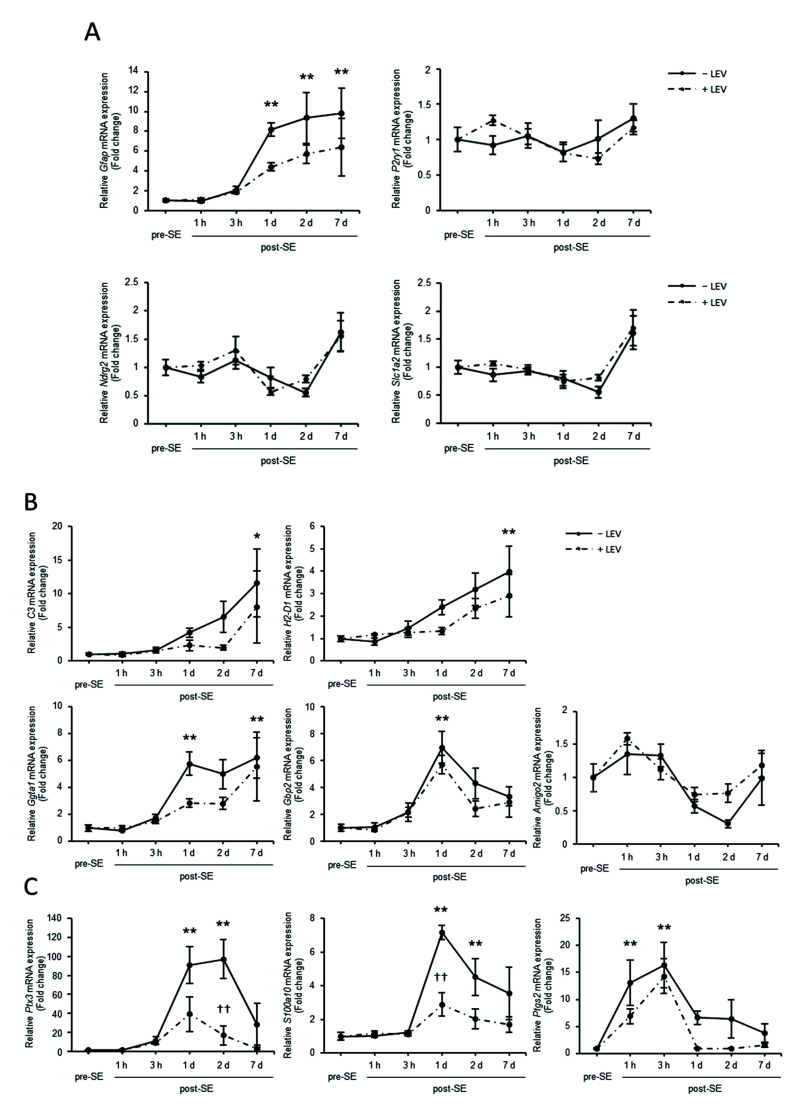
Effect of PILO-SE and LEV administration on the transformation of A1/A2 reactive astrocytes. Time-course real-time PCR analysis of A1-specific mRNA expression during epileptogenesis. Total RNA was extracted from the hippocampi of mice treated with LEV or that were untreated before (pre-) and after (post-) SE at 1 h, 3 h, 1 day, 2 days and 7 days. The mRNA expression levels of astrocytic markers (*Gfap*, *P2ry1*, *Ndrg2* and *Slc1a2*) (**A**), A1-specific markers (*C3*, *H2-D1*, *Ggta1*, *Gbp2* and *Amigo2*) (**B**) and A2-specific markers (*Ptx3*, *S100a10* and *Ptgs2*) (**C**) during epileptogenesis were assessed using real-time PCR. The values were normalized relative to the β-actin (*Actb*) mRNA level and then divided by those of the control group (pre-SE) to calculate the relative mRNA levels. All data points represent the mean ± SEM of at least three independent experiments (*n* = 3–9). Data were analyzed for significant differences by one-way ANOVA and Tukey’s post-hoc test. Results with *p* values < 0.05 were considered statistically significant. ** *p* < 0.01, * *p* < 0.05 vs. pre-SE mice, †† *p* < 0.01 vs. without post-SE mice.

**Table 1 ijms-23-07608-t001:** Primer sequences used for real-time PCR.

Gene	Forward Primer	Reverse Primer
*Il6*	GTCGGAGGCTTAATTACACATGTTC	AATCAGAATTGCCATTGCACAA
*IL1* *rn*	TTGTGCCAAGTCTGGAGATG	CTCAGAGCGGATGAAGGTAAAG
*Il33*	TCCTTGCTTGGCAGTATCCA	TGCTCAATGTGTCAACAGACG
*Il23a*	GACCCACAAGGACTCAAGGAC	ATGGGGCTATCAGGGAGTAGAG
*Ccl2*	GGCTCAGCCAGATGCAGTTAA	CCTACTCATTGGGATCATCTTGCT
*Ccl5*	GCCCACGTCAAGGAGTATTTCTA	ACACACTTGGCGGTTCCTTC
*Ccl6*	ATCAAGCCGGGCATCATCTTTA	TGCCCTCCTTCTCAAGCAAT
*Ccl11*	GAATCACCAACAACAGATGCAC	TCCTGGACCCACTTCTTCTT
*Ccl12*	CATCAGTCCTCAGGTATTGGC	TTGTGATTCTCCTGTAGCTCTTC
*Ccl22*	TGGTGCCAATGTGGAAGACA	GGCAGGATTTTGAGGTCCAGA
*Cx3cl1*	CGCGTTCTTCCATTTGTGTA	CATGATTTCGCATTTCGTCA
*Cxcl1*	ACTGCACCCAAACCGAAGTC	CAAGGGAGCTTCAGGGTCAA
*Cxcl10*	AAGTGCTGCCGTCATTTTCT	GTGGCAATGATCTCAACACG
*Cxcl11*	ATGGCAGAGATCGAGAAAGC	TGCATTATGAGGCGAGCTTG
*Cxcl13*	AGATCGGATTCAAGTTACGCC	TTTGGCACGAGGATTCACACA
*Cxcr3*	AACGTCAAGTGCTAGATGCCT	TCTCGTTTTCCCCATAATCG
*Fosl1*	AGGGCATGTACCGAGACTA	GTGGAACTTCTGCTGCTGG
*Fos*	CCCATCCTTACGGACTCCC	GAGATAGCTGCTCTACTTTGCC
*Tmem119*	ACCCAGAGCTGGTTCCATAG	CGGCTACATCCTCCAGGAAG
*Grin1*	ACTCCCAACGACCACTTCAC	GTAGACGCGCATCATCTCAA
*Slc1a2*	GGTCATCTTGGATGGAGGTC	ATACTGGCTGCACCAATGC
*Cdh5*	TGGCCAAAGACCCTGACAA	TTCGGAAGAATTGGCCTCTGT
*Gfap*	ACCAGCTTACGGCCAACAGT	CCGAGGTCCTGTGCAAAGTT
*P2ry1*	GGCAGGCTCAAGAAGAAGAAT	TCCCAGTGCCAGAGTAGAAGA
*Ndrg2*	ACACCTTATGGCTCGGTCAC	TCTCTTGCATATCCCCGAAC
*C3*	GCAGACCTTAGCGACCAAGT	CCGCAATGACTGTTGGTGTC
*H2-D1*	TCCGAGATTGTAAAGCGTGAAGA	ACAGGGCAGTGCAGGGATAG
*Ggta1*	GTGAACAGCATGAGGGGTTT	GTTTTGTTGCCTCTGGGTGT
*Gbp2*	CAGCTGCACTATGTGACGGA	AGCCCACAAAGTTAGCGGAA
*Amigo2*	CCGATAACAGGCTGCTGGAG	AGAATATACCCCGGCGTCCT
*Ptx3*	CTGCCCGCAGGTTGTGAAA	AGCTTCATTGGTCTCACAGGA
*S100a10*	CATGATGCTTACGTTTCACAGGTT	TGGTCCAGGTCCTTCATTATTTTG
*Ptgs2*	GGGAGTCTGGAACATTGTGAA	GTGCACATTGTAAGTAGGTGGACT
*Egr1*	AGCAGCGCCTTCAATCCTCA	GTCGTTTGGCTGGGATAACT

## Data Availability

The CAGE-seq data have been deposited in the Gene Expression Omnibus (GEO) database (accession number GSE205373). All other data are included in the article.

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
