# Peer review of "Regulation of Inflammation-Related Genes through Fosl1 Suppression in a Levetiracetam-Treated Pilocarpine-Induced Status Epilepticus Mouse Model"

_ijms, 2022, doi:10.3390/ijms23147608_

Round 1

Reviewer 1 Report

Dear Editor

The manuscript by Komori et al investigates the protective effects and the possible mechanisms of levetiracetam action in a pilocarpine –induced status epilepticus mouse model.

The design of the study and the technical quality of the work look somehow convincing and results can be of general interest.

Reviewer 2 Report

Levetiracetam (LEV) is a well-established second-generation antiepileptic drug (AED) and a candidate anti-epileptogenic drug. The authors previously reported that treatment with levetiracetam (LEV) in pilocarpine (PILO) mice model clearly attenuated spontaneous recurrent seizures. Furthermore, they found that LEV protects against BBB failure associated with angiogenesis and brain inflammation as well as abnormal activation of microglia during epileptogenesis. Recently, they identified FosL1 (a marker of neuronal activation) as a new target molecule of LEV showing that it plays a crucial role in the abnormal BV-2 microglial activation and inflammatory response thus preventing the onset of epilepsy. In the same work, they reported an increase in the expression of FosL1 in the hippocampus of PILO-SE mouse model during epileptogenesis. To investigate the early events of epileptogenesis, Komori et al. studied gene expression profiling in the hippocampus of LEV-treated and untreated PILO-SE mice. The dynamics of hippocampal Fosl1 expression was investigated in detail after SE to explore the mechanism underlying the suppression of the inflammatory events that occurs after LEV administration. As a key finding, the authors reported that LEV treatment negatively affects the SE-dependent increase in neurotoxic/proinflammatory A2-astrocyte population via Fosl1 pathway, leaving the neuroprotective A1-phenotype unaffected.

In general, the study was well performed. Solid experimental data are well presented in Figures and Results. The Discussion is in line and well supported by the presented data. The methods applied seems appropriate. The manuscript is well written and clearly organized. The overall data are interesting and certainly contribute to a better understanding the anti-inflammatory mechanism promoted by LEV widely used in epilepsy treatment.

I have few minor comments:

1. In my opinion, the authors should detail the three epilepsy phases experienced by the PILO-mice model in order to elucidate the efficacy of LEV related to the epilepsy phase.

2. In 2.2 section, the comparative study between the expression of FosL1 and Fos genes needs to be better explained. The authors should remark their potential involvement in each epilepsy phase. The sentence “Fosl1 has a completely different expression pattern from that of Fos” need to be argued.  

3. The authors should better describe the statistical analysis. Performing at least three independent experimental sets is mandatory. Furthermore, the distinction between the number of independent experimental sets and the number of technical replicates for each experimental set must be clearly stated.  Numbers (N) of independent animals should be indicated in figure bars or text.

Round 2

Reviewer 2 Report

The authors have certainly improved the manuscript by inserting the suggested revisions which give robustness to the results described.